# Comparison between bupivacaine-lidocaine, dexamethasone mixture and bupivacaine alone for motor recovery after axillary brachial plexus block in distal radius surgery: A prospective randomized trial

**Rawee Jongkongkawutthi, Paweenus Rungwattanakit, Pathom Halilamien, Suppachai Poolsuppasit, Busara Sirivanasandha** *

Department of Anesthesiology, Faculty of Medicine Siriraj Hospital, Mahidol University, Bangkok, Thailand

* busarasiri@gmail.com

## Abstract

### Background

Prolonged motor block, known as "dead arm," which can cause patient discomfort and anxiety, is a serious concern that is often overlooked in ambulatory surgery, particularly in elderly patients. The purpose of this study was to examine the recovery time of motor blockade with bupivacaine and a mixture of bupivacaine-lidocaine-dexamethasone in axillary brachial plexus block.

### Methods

A prospective, randomized, double-blinded controlled trial was conducted with 70 patients scheduled for distal end radius fixation under axillary brachial plexus block. A local anesthetic mixture group (LA-mixture group) received a 21 ml mixture of 0.2% bupivacaine with 1.2% lidocaine and 5 mg of dexamethasone (n = 35). A bupivacaine group received 20 ml of 0.5% bupivacaine with 1 ml of normal saline (n = 35). The primary outcome was the duration of the motor blockade. Secondary outcomes included the duration of sensory blockade, postoperative pain score, and the incidence of rebound pain.

### Results

The demographic data were similar between the two groups. The mean times for recovery of hand grips and sensation were 13.5 ± 7.3 and 12.6 ± 6.2 hours in the LA-mixture group and 15.3 ± 6.7 and 14.6 ± 6.2 hours in the bupivacaine group. Pain scores were not significantly different between the two groups, but the incidence of rebound pain was lower in the LA-mixture group (8.6% and 28.6%, p = 0.031).

**Data availability statement:** All relevant data are within the manuscript.

**Funding:** The author(s) received no specific funding for this work.

**Competing interests:** The authors have declared that no competing interests exist.

## Conclusion

The bupivacaine-lidocaine, dexamethasone mixture failed to enhance motor recovery compared to 0.5% bupivacaine alone. However, patients in the mixture group appeared to experience a lower incidence of rebound pain.

## Trial registration

Thai Clinical Trials Registry TCTR20200114003

## Background

Regional anesthesia is commonly performed for ambulatory hand surgery. Existing data show that regional anesthesia offers several advantages compared with general anesthesia during recovery after ambulatory hand surgery, including reduced opioid and antiemetic consumption and a shortened stay in the postanesthesia care unit [1–4].

The unpleasant feeling after a brachial plexus block is known as a "dead arm" or "prolonged motor block." Patients, especially those undergoing ambulatory procedures, experience difficulties performing routine activities involving their arms below the shoulder for an extended period when using long-acting local anesthetics. This consequence may impact the balance of elderly patients and increase the risk of falling, particularly in those with a history of fractured distal end of the radius due to falls [5–7]. Selecting appropriate local anesthetics for peripheral nerve blocks is crucial.

Bupivacaine is an effective local anesthetic for both central and peripheral nerve blocks, particularly in areas where ropivacaine is inaccessible. Although 0.5% bupivacaine is frequently used to provide postoperative analgesia due to its prolonged duration of action, it may not be suitable for ambulatory surgery. Several studies investigated the use of 0.25% bupivacaine in brachial plexus blocks, often comparing its efficacy and duration to 0.5% bupivacaine [8–10]. The 0.25% bupivacaine provides an effective, albeit less dense, slow onset, and shorter-lasting motor block than higher concentrations [9,10]. It is frequently used in combination with adjuvants to improve block onset, duration, and efficacy. The concentration and combination should be customized to the clinical setting, taking into account aspects such as desired onset time, analgesia duration, and patient safety. While 0.125% bupivacaine is routinely used in postoperative continuous infusions for preserving motor function [11]. On the other hand, 2% lidocaine has a short onset and works effectively for intraoperative use, but it may not be adequate for extended operations. The previous study discovered that, with the bupivacaine-lidocaine mixture had a faster onset time, and the postoperative analgesia was shorter than 0.5% bupivacaine alone [12,13].

Rebound pain is an additional complication that is important in ambulatory surgery. Noncompliant bridging therapy is believed to be the leading cause of rebound pain after the peripheral nerve block subsides, especially in cases of extensive and dense blocks that increase the likelihood of a "dead arm" and rebound pain [7]. Dexamethasone is a commonly used adjuvant that provides prolonged analgesia while reducing rebound pain [14–16].

To achieve a rapid onset and sustained duration of anesthesia suitable for surgery, we prepared a mixture of 1.2% lidocaine and 0.2% bupivacaine, with 5 mg of dexamethasone added as an adjuvant. The total volume of the mixture was 21 milliliters. This combination was intended to promote motor recovery, reduce rebound pain, and extend postoperative analgesia.

Our objective was to compare a mixture of bupivacaine-lidocaine-dexamethasone versus 0.5% bupivacaine with normal saline. We evaluated the motor block duration as the primary

outcome, as well as the sensation duration, time to moderate pain, numeric pain rating scale, and patient satisfaction score. We hypothesized that the local anesthetic mixture would result in a shorter motor block duration than bupivacaine alone for patients undergoing ultrasound-guided axillary brachial plexus block.

## Materials and methods

### Study design and population

This study was a single-center, prospective, double-blind, randomized controlled trial. It was approved by the Siriraj Institutional Review Board (approval number Si265/2020) and registered at the Thai Clinical Trials Registry (TCTR ID: TCTR20200114003), registered 10 January 2020, https://www.thaiclinicaltrials.org/show/TCTR20200114003. The study was conducted on patients aged 20–85 years with American Society of Anesthesiologists physical status I–III who were scheduled to undergo elective open reduction and internal fixation of the distal radius at Siriraj Hospital between May 2020 and August 2021. The inclusion criteria for this study were patients with acute closed distal radius fractures who were undergoing an operation under axillary brachial plexus blockade. The following were the exclusion criteria:

- Body weight of less than 35 kg.

- History of chronic kidney disease with an estimated glomerular filtration rate of less than 50 ml/min/1.73 m2.

- Uncontrolled diabetes with a hemoglobin A1C level greater than 7.5%.

- Contraindication to regional anesthesia.

- Cognitive impairment.

- Preexisting neuropathy.

- Neurological deficit in the upper extremities.

- Known allergies to study medications.

Patients who declined to participate were also excluded from the study.

### Randomization and blinding

A total of 70 patients were recruited and randomized into two groups: a local anesthetic mixture group (LA-mixture group) and a bupivacaine group. Randomization was performed in blocks of four using a computerized random number generator in a block of 4 (www.random-ization.com), and the results were kept in opaque envelopes. Patients in the LA-mixture group (n = 35) received 21 ml of 1.2% lidocaine with 0.2% bupivacaine and dexamethasone 5 mg (12 ml of 2% lidocaine, 8 ml of 0.5% bupivacaine, and 1 ml of dexamethasone) for the axillary brachial plexus block. Patients in the bupivacaine group (n = 35) were administered 20 ml of 0.5% bupivacaine with 1 ml of normal saline solution (total volume of 21 ml). The patients and the assessor were blinded to the treatment allocations.

### Anesthetic protocol

This study was conducted in both inpatient and outpatient settings. Before the operative date, eligible patients were briefed about the research objectives and experimental methods via telephone, and they provided written consent on the day of their operation. Standard monitoring (pulse oximetry, noninvasive blood pressure monitoring, and

electrocardiogram) was performed, and intravenous access was established in the nonsurgical upper extremity. Supplemental oxygen was administered via a nasal cannula at a rate of 2–3 L/min. Sedation was induced using intravenous midazolam (0.01–0.02 mg/kg) and/or fentanyl (0.5–1 mcg/kg).

For both study groups, an anesthesiologist with expertise in regional anesthesia for more than five years performed an ultrasound-guided axillary brachial plexus block. Patients were positioned supine with the shoulder abducted and the elbow flexed in the pre-anesthetic block area. The axilla was sterilized, and an ultrasound probe was used to visualize the axillary artery in a short-axis view. A lidocaine skin wheal was raised, followed by the injection of the local anesthetic using an in-plane ultrasound technique. In the LA-mixture group, a total of 21 ml of 1.2% lidocaine with 0.2% bupivacaine and dexamethasone 5 mg was injected (12 ml of 2% lidocaine, 8 ml of 0.5% bupivacaine, and 1 ml of dexamethasone). In the bupivacaine group, 20 ml of 0.5% bupivacaine was injected with 1 ml of normal saline solution (total: 21 ml).

In the operating room, once the cast was removed from the fractured arm, the block's efficacy was validated by a decrease in cold and pin-prick sensation along the distribution of the radial, medial, ulnar, and musculocutaneous nerves compared with the contralateral arm. Motor blockade was evaluated by an inability to perform both the hand grip and elbow flexion. Patients received an intravenous propofol infusion (50–100 mcg/kg/min) to provide mild to moderate sedation and relieve the ischemic pain due to tourniquet pressure throughout the procedure. In the event of failed regional anesthesia, general anesthesia with a laryngeal mask airway was performed.

During the postoperative period, all patients followed the standard oral postoperative regimen. The regimen included paracetamol (500 mg) or paracetamol with codeine (500/8mg) (1 tablet, every 6 hours as needed); nonsteroidal anti-inflammatory drugs such as ibuprofen (400 mg, every 8 hours), celecoxib (200 mg, every 12 hours), or etoricoxib (90 mg, once daily); aescin (Reparil, 1 tablet three times per day); and tramadol or intravenous morphine for pain relief as needed.

## Outcomes and measurements

The primary outcome was the duration of the motor block. This is measured by the time the block is completed until the patient feels that motor power has returned to preoperative levels. If the blocking effect disappears during sleep, consider the recovery of the motor block at the time they wake up.

The secondary outcomes were 1) the time to the first pain- when the patient feels pain in the surgery area for the first time, 2) worst pain scores, 3) average pain scores, 4) pain medication usage, 5) patient satisfaction scores, 6) desired speed of motor recovery- the amount of time a patient needs to recover their motor block after surgery, and 7) the rebound pain- the pain score increased more than 2 points from the average pain score [17].

The study was planned to include ambulatory and inpatient cases. In their blocked diary, we asked the patients to note the time when they first regained hand movement (time to motor recovery), how long the sensory block lasted, or when they first experienced moderate pain (numeric rating scale ≥ 4) at the surgical site. A single-blinded assessor, a research team member, contacted the patients on the evening of the surgery and again the following evening to collect data and check for complications. This standardized approach ensured consistent questioning and interpretation including 1) the time to motor recovery, 2) the time to the first pain, 3) worst & average pain scores in the first 24 hours, 4) pain medication usage, 5) patient satisfaction scores, 6) desired speed of motor recovery, and 7) the rebound pain.

## Sample size

Prior research on the duration of motor recovery and the brachial plexus block found that the duration of a motor block from 0.5% bupivacaine was 14 ± 5.55 hours [18,19]. Additionally, our pilot study revealed a mean difference of 5 hours (standard deviation of 6 hours) in motor recovery between 0.5% bupivacaine and an LA-mixture groups. Our power analysis indicated that, with an anticipated 10% dropout rate, 32 patients per group (using nQuery Advisor 7.0 program) would be required to demonstrate a statistically significant 5-hour difference in motor power recovery time. The power analysis was based on a 90% power to detect a 5-hour difference with an alpha of 0.05. In total, the study aimed to enroll 70 patients.

## Statistical analysis

All statistical analyses were conducted using IBM SPSS Statistics, version 29 (IBM Corp, Armonk, NY, USA). Data normality was assessed using histograms. Continuous data are presented as the means with standard deviations (SDs) or medians with interquartile ranges [IQRs]. Categorical data are reported as counts and percentages. Comparisons between the two groups utilized the chi-square test for grouped data, the unpaired t-test for normally distributed continuous data, and the Mann–Whitney U test for non-normally distributed continuous data. The statistical significance was defined as a p-value < 0.05.

## Results

Between May 2020 and August 2021, 98 patients with closed distal radius fractures who met the criteria were enrolled. Eighteen patients (18.4%) were excluded for various reasons: eight had concomitant injuries; four displayed glomerular filtration rates below 50 mL/min/1.73 m²; two had hemoglobin A1C levels exceeding 7.5%; two were allergic to an analgesic as per the protocol; one presented acute coronary symptoms presurgery; and one showed signs of median nerve compression. An additional ten patients (10.2%) were excluded due to the unavailability of research staff, resulting in a final sample of 70. No patients dropped out after randomization (Fig 1).

There were no statistically significant differences between the LA-mixture group and the bupivacaine group in terms of patient demographics, the interval between injury and surgery, or preoperative pain scores (Table 1). Of the surgeries, 32 (45.7%) were conducted as ambulatory procedures, and 32 (45.7%) involved the dominant hand. The distribution of these factors was similar between the two groups.

Operation duration and tourniquet periods were comparable between the groups. The choice of intraoperative sedative drugs was consistent between groups, except for fentanyl dosage. The LA-mixture group had a mean intraoperative fentanyl dosage of 0.86 ± 0.26 mcg/kg compared to the bupivacaine group's 1.08 ± 0.4 mcg/kg (p = 0.034), as indicated in Table 2.

One patient in the bupivacaine group had to be converted to a general anesthetic with a laryngeal mask airway during the bone fixation stage due to pain and movement. For this patient, the total administered amounts of propofol, midazolam, and fentanyl were 110 mg, 1.5 mg, and 100 mcg, respectively. However, after the operation, a postoperative reassessment revealed a complete sensory and motor block, negating the necessity for study dropout. The duration of postoperative motor and sensory block recovery for this patient, who had general anesthesia, was also documented.

In the postoperative period (Table 3), no significant differences were observed between the two groups in terms of motor and sensory recovery. For the LA-mixture group, recovery times for hand grip and elbow flexion were 14.14 ± 6.92 hours and 13.19 ± 6.87 hours, respectively.

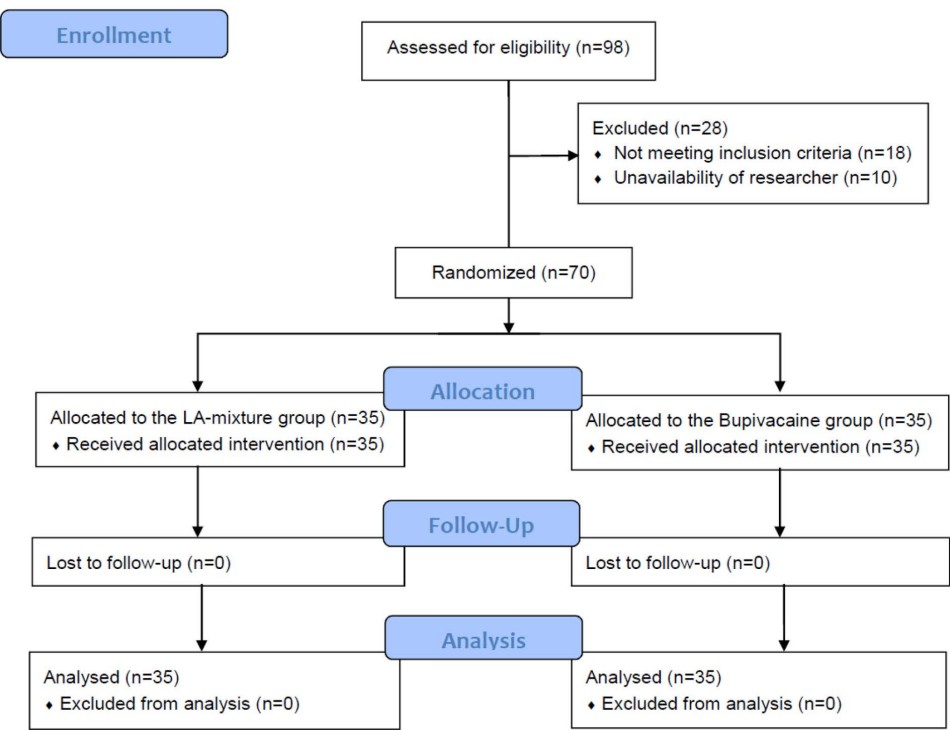

**Fig 1. CONSORT 2010 Flow Diagram.**

**Table 1. Demographic data.**

| | LA-mixture group (n = 35) | Bupivacaine group (n = 35) | p |
|---|---|---|---|
| Age (years) | 54.6 (±16.2) | 58.1 (±15.2) | 0.348 |
| Female gender (n) | 21 (60%) | 27 (77.1%) | 0.122 |
| BMI (kg/m²) | 23.62 (±4.53) | 23.65 (±4.15) | 0.979 |
| ASA classification (n) | | | 0.560 |
| I | 17 (48.6%) | 15 (42.9%) | |
| II | 15 (42.9%) | 14 (40.0%) | |
| III | 3 (8.6%) | 6 (17.1%) | |
| Underlying diseases (n) | | | |
| DM | 2 (5.7%) | 6 (17.1%) | 0.259 |
| HT | 10 (28.6%) | 10 (28.6%) | 1.000 |
| CAD | 0 | 1 (2.9%) | 1.000 |
| CVA | 0 | 3 (8.6%) | 0.239 |
| Other | 17 (48.6%) | 18 (51.4%) | 0.811 |
| GFR (mL/min/1.73 m²) | 88.61 (±23.77) | 90.09 (±16.66) | 0.766 |
| Time to surgery (days) | 11 [9–16] | 9 [6–13] | 0.069 |
| Preoperative pain score (NRS 0–10) | 0 [0–2] | 1 [1–2.5] | 0.275 |
| Ambulatory surgery (n) | 17 (48.6%) | 15 (42.9%) | 0.631 |
| Operation on dominant hand (n) | 14 (40.0%) | 18 (51.4%) | 0.332 |

ASA, American Society of Anesthesiologists; BMI, body mass index; CAD, coronary artery disease; CVA, cerebrovascular accident; DM, diabetes mellitus; GFR, glomerular filtration rate; HT, hypertension; NRS, numeric rating scale.

Data are shown as the means ( ± SD), n (%), and median [interquartile range; IQR].

*p < 0.05 indicates statistical significance.

**Table 2. Intraoperative data.**

|  | LA-mixture group (n = 35) | Bupivacaine group (n = 35) | p |
|---|---|---|---|
| Propofol (mg/kg/min) | 61.04 (±28.21) | 57.02 (±27.99) | 0.552 |
| Fentanyl (mcg/kg) | 0.86 (±0.26) | 1.08 (±0.40) | 0.034* |
| Midazolam (mg) | 0 [0–0] | 0 [0–1] | 0.077 |
| Tourniquet time (min) | 87.77 (±27.04) | 85.94 (±24.12) | 0.766 |
| Operative time (min) | 94.66 (±23.96) | 90.34 (±25.18) | 0.465 |

Data are shown as the means (± SD) and median [IQR].

*p < 0.05 indicates statistical significance.

**Table 3. Postoperative outcomes.**

|  | LA-mixture group (n = 35) | Bupivacaine group (n = 35) | p |
|---|---|---|---|
| Motor recovery (hours) |  |  |  |
| Hand | 14.14 (±6.92) | 15.31 (±6.47) | 0.467 |
| Elbow | 13.19 (±6.87) | 13.54 (±6.68) | 0.833 |
| Time to first pain (hours) | 12.69 (±6.18) | 14.61 (±6.15) | 0.766 |
| Pain score (NRS 0–10) |  |  |  |
| Average pain | 3 [2–5] | 4 [2–5] | 0.261 |
| Worst pain | 5 [3–7] | 5 [3–8] | 0.092 |
| Rebound pain (n) | 3 (8.6%) | 10 (28.6%) | 0.031* |
| Satisfaction score (NRS 0–10) |  |  |  |
| Motor block | 8.00 (±1.70) | 7.83 (±1.34) | 0.641 |
| Sensory block | 8.29 (±1.79) | 8.17 (±1.38) | 0.766 |
| Overall | 8.46 (±1.63) | 8.23 (±1.31) | 0.520 |
| Desired speed of motor recovery (hours) | 0 [0–0] | 0 [0–1] | 0.418 |
| Block wear-off during sleep (n) | 12 (34.3%) | 18 (51.4%) | 0.147 |

NRS, numeric rating scale.

Data are shown as the means (± SD) and median [IQR], n (%).

*p < 0.05 indicates statistical significance.

In contrast, the bupivacaine group exhibited recovery times of 15.31 ± 6.47 hours and 13.54 ± 6.68 hours (p = 0.467 and 0.833, respectively). The 95% confidence interval was -4.37 to 2.02.

Neither group showed significant differences in either average or worst pain scores. In the LA-mixture group, the median average pain score was 3 [IQR 2–5], and the worst pain score was 5 [IQR 3–7]. In the bupivacaine group, the corresponding scores were 4 [IQR 2–5] and 5 [IQR 3–8].

The LA-mixture group had a significantly lower incidence of rebound pain: only 3 patients (8.6%), compared to 10 patients (28.6%) in the bupivacaine group (p = 0.031). All patients expressed a desire to regain hand movement as soon as possible after surgery. Patient satisfaction scores were 8.46 ± 1.63 in the LA-mixture group and 8.23 ± 1.31 in the bupivacaine group (p = 0.520). For the patient who needed general anesthesia, motor recovery times for the hand and elbow were about 12.40 and 2.40 hours, respectively. The time to first pain for this patient was about 16 hours.

Regarding postoperative medication, there was no difference between the groups in terms of the number of patients who received paracetamol, nonsteroidal anti-inflammatory drugs, aescin (Reparil), paracetamol with codeine, tramadol, and intravenous morphine for severe pain after surgery (Table 4).

**Table 4. Postoperative medications.**

|  | LA-mixture group (n = 35) | Bupivacaine group (n = 35) | p |
|---|---|---|---|
| Paracetamol | 28 (80.0%) | 31 (88.6%) | 0.324 |
| NSAIDs | 34 (97.1%) | 33 (94.3%) | 1.000 |
| Aescin (Reparil) | 20 (57.1%) | 27 (77.1%) | 0.075 |
| Paracetamol with codeine | 5 (14.3%) | 11 (31.4%) | 0.088 |
| Tramadol | 8 (22.9%) | 3 (8.6%) | 0.101 |
| Morphine | 6 (17.1%) | 10 (28.6%) | 0.255 |

NSAIDs, nonsteroidal anti-inflammatory drugs.

Data are shown as n (%).

*p < 0.05 indicates statistical significance.

## Discussion

In this randomized trial, we investigated the duration of the motor block after an axillary brachial plexus block under ultrasound guidance. We compared two groups: 1.2% lidocaine and 0.2% bupivacaine with dexamethasone 5 mg totaling 21 ml (the LA-mixture group) versus 0.5% bupivacaine with normal saline solution totaling 21 ml (the bupivacaine group). Our study found no statistically significant differences between the two groups in motor block duration, sensory block duration, or postoperative pain scores. The LA-mixture group showed a mean motor block recovery time of 14.1 hours for hand grip and 13.2 hours for elbow flexion. The bupivacaine group recorded longer mean recovery times of 15.3 hours for hand grip and 13.5 hours for elbow flexion (p = 0.467 and 0.833, respectively).

Several factors may account for these nonsignificant results. First, perineural dexamethasone's effect on motor blockade needs to be considered. Although the primary local anesthetic in the LA-mixture group was lidocaine, which typically wears off in 2–3 hours, adding dexamethasone may extend this period to 5 hours [20,21]. Furthermore, 0.2% bupivacaine predominantly affects sensory function or pain control [22]. However, in our study, adding 5 mg of perineural dexamethasone improved blocking efficiency, extending the motor block duration to 6–8 hours. A study by Aliste et al. reported that the addition of perineural dexamethasone to a mixture of 1% lidocaine and 0.25% bupivacaine increased block duration/analgesia by 4.0–4.7 hours, resulting in a total motor block time of 17.5 hours [16]. This extended motor block duration may be attributed to the synergistic effects of dexamethasone combined with lidocaine and bupivacaine, especially considering the higher concentration of bupivacaine used in their study compared to ours.

Dexamethasone is administered both intravenously and perineurally for pain management. Numerous studies have ascertained that dexamethasone can mitigate pain and prolong sensory block [23–26]. However, the impact of a prolonged motor block has recently gained more attention. Consistent with the findings from several previous studies [16,20,27], our investigation found that dexamethasone increased the duration of the motor block when combined with 1.2% lidocaine. Nevertheless, this effect had not been adequately considered by previous researchers. We recommend that caution should be exercised when adding perineural dexamethasone to peripheral nerve blocks in the lower limb, especially for procedures that require early ambulation, such as femoral nerve block or popliteal nerve block. This is because the motor block effects may persist longer than anticipated and increase the risk of falls.

The tool used for evaluation was another factor that could have influenced the primary outcome. In this study, patient-reported outcomes were deemed appropriate for an ambulatory setting. Moreover, since the focus was on capturing patients' real-life feelings and

opinions regarding the block, the precise duration of the block was not a critical factor, unlike in other studies. However, it is important to note that the accuracy of patient reports in pain and movement diaries can vary; even a feeling of numbness may be misinterpreted as weakness. To mitigate this potential source of error, we ensured that only one team member explained the assessment process before surgery and followed up with consistent post-surgery questions. Future trials should consider incorporating objective measurement tools, such as a hand grip dynamometer to measure pre- and postoperative muscle strength, or evaluating functional capabilities through activities such as grabbing a ball or demonstrating hand movements.

A third factor contributing to the inconsistencies observed was the wearing off of the blocking effect during sleep. In the LA-mixture group, 12 cases (34%) experienced block wear-off during sleep, compared to 18 cases (51%) in the bupivacaine group. This discrepancy could markedly distort the apparent motor block duration. It is worth noting that no patient woke up in the middle of the night in severe discomfort, as all patients were advised to take painkillers before bedtime.

Most participants in our study desired to regain motor function as quickly as possible after surgery. Fredrickson et al. [22] highlighted that the discomfort associated with an insensate arm often required a temporary halt in the continuous infusion of a high-concentration local anesthetic, even though the purpose of the anesthetic was to alleviate pain. While there were no significant differences in motor block duration between groups in our study, refraining from using dexamethasone and reducing the bupivacaine concentration to decrease motor block time might benefit ambulatory surgery patients by enabling them to return to routine activities more swiftly. It is worth noting that most participants reported high satisfaction levels. This is likely due to participants being informed beforehand about the possibility of prolonged motor block and instructed on arm care, reducing potential surprise or frustration about this side effect.

Other anesthesia techniques for distal end radius fixation include general anesthesia, intravenous regional anesthesia, the Bier block, and the wide-awake local anesthesia no-tourniquet (WALANT) block. The WALANT technique has recently gained traction as a safe and effective anesthetic option for various hand surgeries. Originally developed for soft tissue surgeries, the WALANT technique has since been extended to include bone procedures. This method allows for earlier intraoperative assessment and functional wrist recovery while eliminating the adverse effects of sedation [28–30]. However, the WALANT approach is contraindicated for patients who cannot undergo the awake procedure.

As anticipated, no significant difference in sensory duration was observed in our study. However, the incidence of rebound pain was significantly lower in the LA-mixture group than in the bupivacaine group (8.6% vs 28.6%, p = 0.031). Given that rebound pain following upper limb blocks with tense and long-acting local anesthetics can be considerable [7], using a lower but appropriate concentration for the procedure and adding dexamethasone have been shown to minimize rebound pain across a spectrum of peripheral nerve blocks [23,25,26].

Due to lidocaine's faster onset than bupivacaine's, combining these local anesthetics may result in a quicker onset of a tense block than pure bupivacaine. If that is the case, it might explain why the mean intraoperative fentanyl consumption in the LA-mixture group was less than that in the bupivacaine group (0.86 ± 0.26 vs 1.08 ± 0.4 mcg/kg; p = 0.034). However, this difference is not clinically significant. Conversely, the mean propofol and midazolam doses for sedation were not significantly different. Notably, one patient in the bupivacaine group necessitated general anesthesia, yet the block proved effective upon reassessment at the end of the procedure. The combination of short- and long-acting local anesthetics may be particularly beneficial in situations that require a rapid onset or adherence to a tight schedule.

However, the onset of the local anesthetics combination may be determined by the concentration of lidocaine in the mixture [31,32].

It is well established that the risk of local anesthetic systemic toxicity is higher with bupivacaine due to its high cardiac toxicity and narrow therapeutic window, which can lead to severe consequences [33]. To curtail these potential adverse side effects, the bupivacaine concentration was scaled down, and the medication was mixed with lidocaine, which has a broader therapeutic window. We demonstrated that the mix of 1.2% lidocaine, 0.2% bupivacaine, and 5 mg of dexamethasone achieved a similar duration of sensorimotor blockade as 0.5% bupivacaine alone while theoretically reducing the risk of local anesthetic toxicity. This mixture may present a safer alternative for blocking techniques requiring high doses or volumes of local anesthesia. However, this study observed no instances of local anesthetic systemic toxicity.

The study had several limitations. First, the research was conducted at a single center, potentially affecting the generalizability of the findings. Second, the ambulatory setting introduced challenges in data collection, especially in the realm of phone-based evaluations. This setting also posed challenges in ensuring patient compliance with postoperative medication and pain management protocols. Nevertheless, these challenges are unlikely to have influenced the primary study outcomes, given that previous research has indicated similar efficacy among oral nonsteroidal anti-inflammatory drugs in perioperative pain reduction [34]. Third, the absence of a group that received a local anesthetic without dexamethasone makes it difficult to attribute observed outcomes exclusively to our chosen interventions. Although this design choice aimed to achieve comparable pain management outcomes, it may have inadvertently led to an underestimation of the influence of dexamethasone on motor blockade. Finally, this study's definition of rebound pain is not universally accepted [17,23,35]. The criterion we adopted for analysis in ambulatory cases was an increase in pain score of more than two points. We acknowledge the need for improved standardization in future studies to enhance reproducibility. Furthermore, future research should prioritize stratifying patients at higher risk of rebound pain to yield more comprehensive and clinically relevant insights.

## Conclusions

In summary, the combination of 1.2% lidocaine, 0.2% bupivacaine, and 5 mg dexamethasone is a viable alternative to 0.5% bupivacaine in ambulatory settings. This mix lessens rebound pain and possibly reduces the required dose of intraoperative fentanyl due to the combination's rapid onset of action. Nevertheless, the duration of the sensorimotor block for the ultrasound-guided axillary brachial plexus block remains the same.

## Supporting information

**S1 File. Raw data.**
(XLSX)

## Acknowledgments

The authors are indebted to Miss Julaporn Pooliam for assistance with the methodology, Mrs. Nichapat Thongkaew, and Miss Chayanan Thanakiattiwibun for their invaluable support with administrative work, and Mr. David Park for thoroughly proofreading the manuscript.

## Author contributions

**Conceptualization:** Rawee Jongkongkawutthi, Busara Sirivanasandha.

**Data curation:** Rawee Jongkongkawutthi, Paweenus Rungwattanakit, Pathom Halilamien, Suppachai Poolsuppasit.

**Formal analysis:** Rawee Jongkongkawutthi, Busara Sirivanasandha.

**Investigation:** Rawee Jongkongkawutthi, Paweenus Rungwattanakit, Pathom Halilamien, Suppachai Poolsuppasit.

**Methodology:** Busara Sirivanasandha.

**Project administration:** Busara Sirivanasandha.

**Supervision:** Busara Sirivanasandha.

**Writing – original draft:** Rawee Jongkongkawutthi, Busara Sirivanasandha.

**Writing – review & editing:** Busara Sirivanasandha.

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
