## [Decision Letter · Decision Letter 0]

29 Dec 2024

PONE-D-24-39685Comparative Motor Recovery in Axillary Brachial Plexus Block for Distal Radius Surgery: A Prospective, Randomized Controlled Trial of Bupivacaine-Lidocaine-Dexamethasone Mixture vs. Bupivacaine AlonePLOS ONE

Dear Dr. Sirivanasandha,

Thank you for submitting your manuscript to PLOS ONE. After careful consideration, we feel that it has merit but does not fully meet PLOS ONE’s publication criteria as it currently stands. Therefore, we invite you to submit a revised version of the manuscript that addresses the points raised during the review process.

**ACADEMIC EDITOR:** Authors are required to reply all the queries, raised by the reviewers.

We look forward to receiving your revised manuscript.

Kind regards,

Priti Chaudhary, M.S.

Academic Editor

PLOS ONE

Additional Editor Comments (if provided):

Reviewers' comments:

Reviewer's Responses to Questions

**Comments to the Author**

1. Is the manuscript technically sound, and do the data support the conclusions?

Reviewer #1: Partly

Reviewer #2: Yes

Reviewer #3: Partly

2. Has the statistical analysis been performed appropriately and rigorously?

Reviewer #1: Yes

Reviewer #2: Yes

Reviewer #3: Yes

3. Have the authors made all data underlying the findings in their manuscript fully available?

Reviewer #1: Yes

Reviewer #2: Yes

Reviewer #3: Yes

4. Is the manuscript presented in an intelligible fashion and written in standard English?

Reviewer #1: Yes

Reviewer #2: Yes

Reviewer #3: Yes

5. Review Comments to the Author

Reviewer #1: This manuscript describes a prospective randomised controlled study on the comparison between the effect of bupivacaine alone or a combination of bupivacaine, lignocaine and dexamethasone on recovery from motor blockade following axillary block for hand surgery. The manuscript is well presented and the original premise is reasonable. However, there are several aspects to this study that render the manuscript not novel or adding significant value to current evidence base.

First, the concept of dense motor block is well known among clinicians who practise these blocks. Using 0.5% bupivacaine is uncommon for this reason and in many centres, the concentration of bupivacaine used is 0.25% and in some instances even 0.125%. It is well established that using low concentrations of bupivacaine reduce the incidence of a dense block. This doesn’t require the multiple drug combinations used in the LA mixture arm of this study, thus obviating the need for this study.

Second, the addition of dexamethasone is not sufficiently motivated in this manuscript. If dexamethasone is used to increase the duration of analgesia in the absence of extended motor block, then there should be another arm of the study that investigated the combination of lignocaine and 0.2% bupivacaine. Clinical experience suggests that motor blockade would have been shorter in duration in this third group. Further, the authors have presented evidence from previous research that dexamethasone can increase duration of motor block. This confuses the primary outcome measure for the sake of assessing reduction in rebound pain (a secondary outcome measure).

Third, rebound pain as described in the study is not defined and measured according to previously established methodology. Hence the findings from this study can not be reproduced in future studies. There is no indication in the manuscript of stratification of patients who may be at higher risk of rebound pain and therefore it is difficult to understand the results presented here.

Fourth, there is some discrepancy between the statistical metrics presented in the table and in the text (rebound pain 3.6% and 8.6%, p values: 0.017 and 0.0017). These indicate either typographical errors or significant errors,

Finally, the statistically significant difference between the dose of fentanyl required in the 2 groups is clinically insignificant. There are also several grammatical and typographical errors that need to be corrected.

Reviewer #2: The sample size and statistical analysis plan are simple and well stated. Presumably the significance level is set at 0.05. Please explain the asterisk in the footnote area of Tables 2 and 3. The outcomes are in Table 3 with the only statistically significant result being a comparison of proportion of rebound pain. All the tests in Table 3 are univariate. This prompts the question of multiple comparisons with a need for p-value adjustment. The authors should acknowledge this or at least tell why no adjustment is made.

The study limitations are outlined appropriately.

Reviewer #3: i appreciate the authors efforts in conducting this trial that investigated the ability of local anesthetic mixture to avoid delayed motor recovery. i have the following comments:

1- although the title has all components of CONSORT checklist, but it is haphazardly written and needs to be arranged in a logical way to avoid confusion

suggestion:

comparison between Bupivacaine- Lidocaine, Dexamethasone Mixture vs. Bupivacaine Alone for motor recovery after Axillary Brachial Plexus Block in Distal Radius surgery; a prospective randomized trial.

2- in abstract:

- it is better to replace ( satisfaction) with the outcome that reported in the results and conclusion ( incidence of pain rebound)

- define the 1ry and 2ry outcomes in methods

- add only the significant p value to the results.

- why conclusion based on the patients’ perspective .. the 1ry outcome as mentioned in the text is the duration of motor recovery assessed objectively.

3- in methods:

-anesthesiologist with expertise in regional anesthesia .. add the number of experience years

- why the motor block was not assessed from the onset of motor block till recovery! including the surgery duration.

- regarding 2ry outcomes

define the time point of measurement for each parameter and the method used for measurement.

also mention which pain medication dose you follow up ?

mention the frequency you followed up the pain scores during a specified duration.

what is the difference between patient satisfaction scores, and the patient’s preferred duration for motor recovery ??

- mention the software used for sample size calculation

- mention the level of significance in the statistical analysis.

4- in the results:

- the table is inserted after the cited text not before it.

- according to table 2 :

Axillary block primarily affects the radial, median, and ulnar nerves, with less effect on the musculocutaneous and axillary nerves. So the Tourniquet sites (usually above the elbow) may not be adequately anesthetized.

the authors should clarify how they deal with this deficiency in the methods ( anaesthesia protocol)

6. PLOS authors have the option to publish the peer review history of their article (what does this mean? ). If published, this will include your full peer review and any attached files.

**Do you want your identity to be public for this peer review?** For information about this choice, including consent withdrawal, please see our Privacy Policy .

Reviewer #1: No

Reviewer #2: No

Reviewer #3: No

---

## [Author Response · Author response to Decision Letter 1]

19 Feb 2025

We have attached a file 'respond to reviewer'.

---

## [Decision Letter · Decision Letter 1]

3 Mar 2025

Comparison Between Bupivacaine-Lidocaine, Dexamethasone Mixture and Bupivacaine Alone for Motor Recovery After Axillary Brachial Plexus Block in Distal Radius Surgery: A Prospective Randomized Trial

PONE-D-24-39685R1

Dear Dr.Busara Sirivanasandha,

We’re pleased to inform you that your manuscript has been judged scientifically suitable for publication and will be formally accepted for publication once it meets all outstanding technical requirements.

Kind regards,

Priti Chaudhary, M.S.

Academic Editor

PLOS ONE

Additional Editor Comments (optional):

Reviewers' comments:

Reviewer's Responses to Questions

**Comments to the Author**

1. If the authors have adequately addressed your comments raised in a previous round of review and you feel that this manuscript is now acceptable for publication, you may indicate that here to bypass the “Comments to the Author” section, enter your conflict of interest statement in the “Confidential to Editor” section, and submit your "Accept" recommendation.

Reviewer #2: (No Response)

2. Is the manuscript technically sound, and do the data support the conclusions?

Reviewer #2: Partly

3. Has the statistical analysis been performed appropriately and rigorously?

Reviewer #2: Yes

4. Have the authors made all data underlying the findings in their manuscript fully available?

Reviewer #2: Yes

5. Is the manuscript presented in an intelligible fashion and written in standard English?

Reviewer #2: Yes

6. Review Comments to the Author

Reviewer #2: I may have confused the author with my question of multiple comparisons. Rebound pain is another dimension, but also another statistical test. That is why some Bonferroni adjustment is usually required in this case as multiple separate p-values are being generated in this table. At least alert the reader that this p-value should be interpreted as not being adjusted.

7. PLOS authors have the option to publish the peer review history of their article (what does this mean? ). If published, this will include your full peer review and any attached files.

**Do you want your identity to be public for this peer review?** For information about this choice, including consent withdrawal, please see our Privacy Policy .

Reviewer #2: No

---

## [Editor Report · Acceptance letter]

PONE-D-24-39685R1

PLOS ONE

Dear Dr. Sirivanasandha,

I'm pleased to inform you that your manuscript has been deemed suitable for publication in PLOS ONE. Congratulations! Your manuscript is now being handed over to our production team.

Kind regards,

on behalf of

Dr. Priti Chaudhary

Academic Editor

PLOS ONE